# Self-Selection of Agricultural By-Products and Food Ingredients by *Tenebrio molitor* (Coleoptera: Tenebrionidae) and Impact on Food Utilization and Nutrient Intake

**DOI:** 10.3390/insects11120827

**Published:** 2020-11-24

**Authors:** Juan A. Morales-Ramos, M. Guadalupe Rojas, Hans C. Kelstrup, Virginia Emery

**Affiliations:** 1Biological Control of Pests Research Unit, National Biological Control Laboratory, USDA-ARS, Stoneville, MS 38776, USA; guadalupe.rojas@usda.gov; 2Beta Hatch Inc., 200 Titchenal Road, Cashmere, WA 98815, USA; hans@betahatch.com (H.C.K.); virginia@betahatch.com (V.E.)

**Keywords:** insects as feed and food, nutrition, food assimilation, food conversion, insect dietetics, insect rearing, macro-nutrients

## Abstract

**Simple Summary:**

Insects have been considered as an alternative to fishmeal in animal feed formulations. Current methods for mass producing them remain expensive and, although cost is not the current market driver for insect products, they remain off the main stream. One way to reduce production costs is to lower the cost of insect diets. This could be accomplished by using agricultural by-products as ingredients to formulate insect diets. In this study 20 ingredients were tested as dietary components for the yellow mealworm. Ingredients included dry potato and cabbage; the bran of wheat and rice; by-product meals from vegetable oil production; spent distiller’s grains from brewery and ethanol production; and hulls of different grains. A method called self-selection was used to approach the optimal proportion of these ingredients in mealworm diets by measuring their relative consumption. Nine combinations of eight ingredients were presented to groups of mealworms while carefully measuring the relative consumption of each ingredient. Results showed that the most suitable ingredients for mealworm production were dry cabbage and potato, the bran of wheat and rice, the meals of canola and sunflower, and distilled grains from corn and barley. This information will be used to formulate and evaluate diet formulations for the yellow mealworm in future research.

**Abstract:**

Nutrient self-selection was used to determine optimal intake ratios of macro-nutrients by *Tenebrio molitor* L. larvae. Self-selection experiments consisted of 9 combinations (treatments) of 8 ingredients, from a total of 20 choices, radially distributed in a multiple-choice arena presented to groups of 100 *T. molitor* larvae (12th–13th instar). Larvae freely selected and feed on the pelletized ingredients for a period of 21 days at 27 °C, 75% RH, and dark conditions. Consumption (g) of each ingredient, larval live weight gained (mg), and frass production were recorded and used to calculate food assimilation and efficiency of conversion of ingested food. The macro-nutrient intake ratios were 0.06 ± 0.03, 0.23 ± 0.01, and 0.71 ± 0.03 for lipid, protein, and carbohydrate, respectively on the best performing treatments. The intake of neutral detergent fiber negatively impacted food assimilation, food conversion and biomass gain. Food assimilation, food conversion, and biomass gain were significantly impacted by the intake of carbohydrate in a positive way. Cabbage, potato, wheat bran, rice bran (whole and defatted), corn dry distillers’ grain, spent brewery dry grain, canola meal and sunflower meal were considered suitable as *T. molitor* diets ingredients based on their relative consumption percentages (over 10%) within treatment.

## 1. Introduction

In recent years, the yellow mealworm, *Tenebrio molitor* L. (Coleoptera: Tenebrionidae) has been considered as a potential source of animal protein in feeds for fish [1,2,3,4,5,6] and livestock [7,8,9,10,11,12,13,14]. An increasing number of companies have been founded every year since 2013 that focus on insect mass production as animal feed [15]. One of the most important aspects of mass production involves the formulation of inexpensive yet effective diets that maximize biomass productivity over time. Recent research has focused on the potential use of agricultural by-products as insect food to reduce production costs of insect biomass [16,17,18,19]. However, current studies on *T. molitor* have focused on the effects of single ingredients on biological and food utilization parameters. No attempts have been made to evaluate combinations of multiple by-products as ingredients with the aim to develop diets for *T. molitor*. Self-selection studies incorporating by-products have been used to develop complete diets for the house cricket, *Acheta domesticus* L. [20].

Developing adequate diet formulations for insects using multiple undefined (oligidic) ingredients is a complex procedure and can take multiple years of research, particularly in insect species with long life cycles such as *T. molitor*. Optimal diets can be obtained from multiple oligidic ingredients by allowing insects to select the optimal ratios of each ingredient in a multiple-choice experimental setting. This method is known as self-selection and was first proposed by Waldbauer and Friedman (1991) [21]. The objectives of this study were to (1) determine the ingredients with the highest potential for formulation of insect diets using the self-selection method, (2) establish the optimal macro-nutrient intake ratios of *T. molitor* based on the self-selected intake of 20 ingredients, and (3) explore the impact of intake of macro-nutrients, neutral detergent fiber (NDF), phytosterol, and minerals including Fe, Mg, Ca, Zn, Cu, and Mn on the biomass gain, food assimilation and efficiency of conversion of ingested food (ECI).

## 2. Materials and Methods

### 2.1. Experimental Design

The colony stock, the rearing procedures and rearing hardware used in this study were as described by Morales-Ramos et al. [22]. Larvae of *T. molitor* used in the experiments were separated by size from the stock colony using sifters of standard numbers 10 and 12, which selected larvae with head capsule width measuring between 1.4 to 2 mm. According to estimates by Morales-Ramos et al. [23] these head capsule measurements correspond to larvae between 3 and 5 instars prior to pupation, which could include instars 11 to 14. However, because the stock used in this study has been selected for larger size, the experimental group of larvae could have included earlier instars.

Experimental units consisted of groups of 100 larvae maintained in multiple-choice arenas designed to provide equal access to 8 different food choices. Groups of larvae from each experimental unit were weighed at the beginning and end of the experiment and their weight was recorded for each of the units of each treatment. The multiple-choice arenas consisted of breathable round plastic dishes (120 × 25 mm, Pioneer Plastics 53C, Pioneer Plastics, North Dixon, KY, USA) modified by the addition of 8 sample plastic vials 20 mL (72 mm height × 25 mm diameter) (Product # 73400, Kartell s. p. a., Noviglio, Milan, Italy). The sample vials were cut to a height of 35 mm to fit inside the dishes and assembled perpendicularly to the dish in a radial pattern (45° apart) with equal distance to the center of the dish (Figure 1A). A 5 mm diameter opening was drilled into one side of each of the vials pointing perpendicularly to the center of the dish, to allow larvae to enter the vials (Figure 1B, a). A depression (90 × 2 mm) was constructed at the center of the dish with screened bottom (0.5 mm screen openings) to allow the collection of frass in a second dish located under the arena (Figure 1).

### 2.2. Food-Choice Treatments

Food choices consisted of 20 food products and agricultural by-products, which included dry white cabbage, potato flour, alfalfa pellets, wheat bran, rice bran (whole and solvent defatted), spelt screenings; meals from canola, soybean, olive, sunflower, cotton, and kelp; hulls from rice, oat, and peanut, and coffee chaff; and dry distilled grains from corn, wheat, and barley from ethanol production and brewery. Nine treatments of eight different combinations of these food choices were selected for the study (Table 1). The criterium of selection for the combination treatments was based on the relative content of each of the macro-nutrients (protein, lipid, and carbohydrate) and dietary fiber as neutral detergent fiber (NDF). Combinations should contain at least one food choice with high content of each of the macro-nutrients and dietary fiber to allow the mealworm larvae to select a complete diet. For instance, diet 1 contains canola meal as high protein ingredient, potato flour as high carbohydrate, corn distilled grain and dry cabbage for lipids and hulls from peanuts and rice as high fiber ingredients (Table 1).

Food ingredients were ground into a fine powder using a high-speed food processor. Powdered food ingredients were individually mixed with reverse osmosis (RO) water at 50% to 70% ratio to obtain a consistency of dough. The food ingredients were then formed into sticks using a cut 10 mL syringe. These sticks were dried in a vacuum oven at 50 °C for a period of 48 h. This procedure resulted in stable dry sticks of each of the food ingredients listed in Table 1 with dimensions that allowed them to be introduced in the compartments of the multiple-choice arenas (Figure 1C).

Food combination treatments consisted of 10 repetitions each (=10 experimental units totaling 1000 larvae). Food ingredients were randomly distributed in the arena compartments to minimize the proximity effects among the different ingredients. At the beginning of the experiment, a measured amount of each of the corresponding food ingredients was added to the corresponding arena compartment in each of the experimental units. The initial amount of each food ingredient provided was recorded for each of the experimental units from each of the combination treatments. Experimental units were maintained in environmental chambers at 27 °C, 75% RH (relative humidity) and dark conditions for a period of three weeks. Experimental units were monitored daily to observe the consumption of each of the ingredients. Food ingredients that were depleted by consumption, were replenished with a measured amount of the corresponding food ingredient, which was recorded for each of the experimental units.

### 2.3. Data Collection and Analysis

At the end of a three-week period, larvae from each experimental unit were counted and weighed alive as a group. The remaining food was collected separately by ingredient, separated from frass, dried in a vacuum oven, and weighed. Frass was separated from food by sifting the remains using a standard No. 35 sieve (0.5 mm openings). The frass was collected, dried, and weighed using the same drying procedure. To collect the remaining food, all the vials in the arena were capped, the arena was inverted, and the contents of each vial were emptied, one by one, into a standard number 35 sieve by removing the cap to separate food from frass. The consumption of each ingredient (I_i_) was calculated as total weight added of ingredient ‘i’—remaining weight of ingredient ‘i’, were i = 1 to 8. The total food consumption (FC) was calculated as the sum of the consumption of all the eight ingredients. Assimilated food (AF) was calculated as AF = FC − frass weight. The percent consumption of each ingredient was calculated as (I_i_/FC) × 100. The weight of live mealworm biomass gained (LWG) was calculated as ending group weight—initial group weight. Mortality was extremely low (0.32 ± 0.21%) and dead larvae were cannibalized by surviving larvae (no cadavers were found), therefore, the ending live biomass measure per group included the loss of biomass due to mortality. Because the initial biomass dry weight could not be directly determined, the dry weight biomass gained (DWG) was calculated as LWG × the proportion of dry matter of mealworm larvae. The proportion of dry weight of mealworm larvae was previously determined from 25 groups of 10 larvae, which were weighed live, then frozen at −25 °C, dried in a vacuum oven at 50 °C, and weighed dry. The dry weight proportion of *T. molitor* late instar larvae was 0.38. The efficiency of conversion of ingested food (ECI) was calculated based on Waldbauer [24] as ECI = (DWG/FC) × 100 for each experimental unit.

Nutrient intake by *T. molitor* larvae was estimated from the self-selected consumption of the choice ingredients using the nutrient matrix calculation described by Morales-Ramos et al. [20,25]. The macro nutrient (lipid, protein and carbohydrate) content of the ingredients used in this study was obtained from data published in multiple sources [26,27,28,29,30,31,32,33,34,35,36,37]. The nutrient intake data was used to calculate the 3-way ratios of macro nutrient intake as described by Morales-Ramos et al. [25] and calculated as protein intake ratio = Pi/MNi, lipid intake ratio = Li/MNi, and carbohydrate intake ratio = Ci/MNi, where Pi, Li, and Ci are intakes of protein lipid and carbohydrate, respectively and MNi is the total intake of all three macronutrients and the sum of all three ratios is always = 1. The intake of other nutrients including neutral detergent fiber (NDF) and minerals including iron, magnesium, manganese, calcium, and zinc was also estimated.

Data consisting of live biomass gained, total food consumption, percent food assimilation, and efficiency of conversion of ingested food were compared among treatments using general linear mixed model (GLMM) and the Tukey–Kramer HSD (honestly significant difference) test for least square means of JMP software version 14.1 [38]. The effect of nutrient intake on food assimilation and efficiency of food conversion (ECI) was analyzed using multiple regression. The stepwise followed by backwards elimination methods were used to determine the optimal number of independent variables required in the model to explain food assimilation and ECI using the *C_p_* statistic as the criterion to include or exclude variables [38,39,40].

## 3. Results

The means of consumption of each of the ingredients within each of the combination treatments are presented in Table 2. The relative consumption of each ingredient within combination treatments is illustrated as percentages in Figure 2. The ingredients that were consumed in higher proportion were dry potato in treatment 8 (41.01%); crude rice bran in treatments 3, 4, 5, 6, and 7 (40.47%, 34.87%, 30.37%, 32.27% and 33.18%, respectively); wheat bran in treatments 1 and 9 (30.49% and 37.1%, respectively); and corn dry distiller’s grain with solubles (DDGS)in treatment 2 (34.63%). The least consumed ingredients were rice hulls in treatments 1 and 2 (0.31% and 0.13%, respectively); coffee chaff in treatment 8 (1.48%); peanut hulls in treatments 3, 6 and 7 (1.66%, 2.77%, and 3.19%, respectively); soybean meal in treatment 9 (2.06%); olive meal in treatment 4 (2.27%); and sunflower meal in treatment 5 (2.58%) (Figure 2). In general, highly consumed ingredients had a high carbohydrate content. The ingredients consumed in low percentages generally contained high amounts of fiber at the expense of other nutrients, such as rice hulls, coffee chaff and peanut hulls or have a combination of high fiber and high protein contents like meals of olive, soybean and sunflower.

Despite the great diversity observed in the relative consumption of ingredients between treatments, the intake ratios of macro nutrients (lipid + protein + carbohydrate = 1) tended to converge close to a set of ranges between 0.03 to 0.16 of lipid, 0.21 to 0.25 of protein and 0.62 to 0.74 of carbohydrate (Table 3, Figure 3) with overall means of 0.1 ± 0.04, 0.24 ± 0.04, and 0.66 ± 0.06 for lipid, protein, and carbohydrate, respectively. The only exception was treatment 2 which showed significantly higher protein (0.36) (*F* = 379.1; df 8, 81; *p* < 0.0001) and lower carbohydrate (0.55) (*F* = 460.8; df 8, 81; *p* < 0.0001) intake ratio than all the other treatments (Table 3) outlying visibly in the graph of Figure 3. However, the rest of the treatments showed some significant differences among them in the macro nutrient intake ratios (*F* = 304, 379.1, and 460.8 for lipid, protein and carbohydrate, respectively; df 8, 81; *p* < 0.0001) that were less obvious in Figure 3 (Table 3).

These differences in the intake ratios of macro nutrients resulted in significant differences in group live biomass gain (*F* = 10.15; df = 8, 81; *p* < 0.0001), overall dry-weight food consumption (*F* = 10.91; df = 8, 81; *p* < 0.0001), food assimilation (*F* = 29.13; df = 8, 81; *p* < 0.0001), and ECI (*F* = 28.41; df = 8, 81; *p* < 0.0001) among choice treatments (Figure 4). The highest live biomass gain was observed in treatment 5 (7.3 ± 0.28 g), followed by treatments 1 (6.91 ± 0.41 g) and 7 (6.83 ± 0.74 g). The highest assimilation was observed in treatment 8 (55.25 ± 2.03%) followed by treatment 5 (50.86 ± 1.86%). The highest ECI was observed in treatment 5 (9.87 ± 0.45%) followed by treatment 8 (9.48 ± 0.64%). In general, the best performing treatments were 5, 8, and 1 (Figure 4). Treatment 2 was the worst performer among choice treatments, showing the lowest live biomass gain (5.45 ± 0.49 g), the lowest food assimilation (39.19 ± 2.11%), and the lowest ECI (7.18 ± 0.5%) (Figure 4). The low performance of larvae groups of treatment 2 may be associated with the significant deviations in macronutrient intake ratios observed in this treatment (Figure 3). The optimal macro-nutrient ratios for *T. molitor* may be closer to those observed in average for treatments 1, 5, and 8, which were 0.06 ± 0.03, 0.23 ± 0.01, and 0.71 ± 0.03 for lipid, protein, and carbohydrate, respectively.

Live biomass gain was significantly impacted by efficiency of food conversion (ECI) (*R*^2^ = 0.53; *F* = 100.74; df = 1, 88; *p* < 0.0001) and food assimilation (*R*^2^ = 0.13; *F* = 13.35; df = 1, 88; *p* = 0.0004) in a positive way. Consumption of some ingredients have significant effects on biomass gain, food assimilation and ECI. For instance, consumption of potato had a significant positive effect on food assimilation (β = 0.01; *R*^2^ = 0.57; *F* = 116.64; df = 1, 88; *p* < 0.0001), but consumption of corn DDGS had the opposite effect on food assimilation (β = −0.007; *R*^2^ = 0.21; *F* = 23.0; df = 1, 88; *p* < 0.0001).

Ingredients that had a mean consumption percentage of at least 10% in any given choice treatment were considered relevant ingredients (RI). Relevant ingredients included potato, cabbage, wheat bran, crude rice bran, defatted rice bran, corn DDGS, spent brewery DG, canola meal, and sunflower meal. Multiple regression analysis indicated that the consumption of all the relevant ingredients had a significant positive effect on live biomass gain (*R*^2^ = 0.7; *F* = 20.75; df = 9, 80; *p* < 0.0001). Only consumption of potato, cabbage, rice bran whole, and spent brewery DG had a significant positive effect on food assimilation (partial *F* Ratios = 49.47, 12.17, 6.62, and 8.55; df = 9, 80; *p* < 0.0001, = 0.0008, = 0.0119, and = 0.0045, respectively). Significant negative effects on food assimilation were observed with consumption of canola and sunflower meals (partial *F* Ratios = 6.39 and 4.49; df 9, 80; *p* = 0.0135 and 0.0371, respectively). The resulting optimized model for assimilation (after stepwise) agreed with the full model analysis including the 6 variables that showed significant effects on food assimilation (*R*^2^ = 0.75; *p* = 42.6; df = 6, 83; *p* < 0.0001). In the full model (9 independent variables) the efficiency of food conversion (ECI) was only affected significantly by the consumption of potato, and this effect was positive (partial *F* Ratio = 13.31; df 9, 80; *p* = 0.0005). However, when this model was analyzed with the stepwise method, an optimized 3-variable model resulted that included potato, rice bran, and canola meal all affecting ECI significantly and positively (*R*^2^ = 0.64; *p* = 49.9; df = 3, 88; *p* < 0.0001). Significant quadratic effects on live biomass gain were observed from consumption of potato (β_1_ = 0.146, β_2_ = −0.023; *R*^2^ = 0.18; *F* = 9.47; df = 2, 87; *p* = 0.0002), corn DDGS (β_1_ = 0.077, β_2_ = −034; *R*^2^ = 0.38; *F* = 26.2; df = 2, 87; *p* < 0.0001) and spent brewery DG (β_1_ = −0.105, β_2_ = −0.024; *R*^2^ = 0.27; *F* = 16.02; df 2, 87; *p* < 0.0001). Biomass gain was maximized at an intermediate level of consumption of these three ingredients.

Intake ratios of some nutrients had a significant impact on food assimilation and efficiency of food conversion (ECI). The optimal multiple regression models obtained after stepwise and backwards elimination procedures consisted of only 2 dependent variables explaining food assimilation and 4 variables explaining ECI. Models are valid only within the ranges observed for these variables, presented in Table 4. Food assimilation was impacted significantly by carbohydrate and neural detergent fiber (*R*^2^ = 0.73; *F* = 117.93; df = 2, 87; *p* < 0.0001) (Table 5). These two variables also impacted ECI in addition to the minerals Mg and Mn (*R*^2^ = 0.73; *F* = 57.48; df = 4, 85; *p* < 0.0001) (Table 6).

## 4. Discussion

It is apparent by the results presented in this study that *T. molitor* larvae tend to balance their intake of macro nutrients by selecting among a variety of ingredients when feeding. This agrees with previous studies confirming the ability of *T. molitor* to self-select for optimal macro-nutrient intake ratios [41,42,43,44]. The intake ratios of macro nutrients by *T. molitor* larvae converged within a narrow range of values among eight of the nine combination treatments of different food ingredients. Treatment 2 was the exception showing excess intake of protein and reduced intake of carbohydrate. Deviation of macro-nutrient intake ratios observed in treatment 2 coincided with a low performance of growth and food utilization of the larvae grown in this treatment. The reason for the deviations in macro-nutrient intake ratios observed in treatment 2 may have been the absence of an additional ingredient with low protein content besides defatted rice bran. There was an unusually high consumption of corn DDGS (34.66 ± 4.13%) and spent brewery DG (29.75 ± 1.46%) in this treatment resulting in a combined mean consumption of 64.41% of these two ingredients from the mean total food consumption in treatment 2. In the other three treatments where these two ingredients were present together (treatments 5, 6, and 7), their combined consumption did not exceed 26% of the total food consumed. Additionally, consumption of corn DDGS and spent brewery DG did not exceed 21.5% when presented alone within the food choices (treatments 1, 3, 4, and 8). The high consumption of these two distilled grain ingredients in treatment 2 is itself an anomaly and may have been driven by the need for lipid intake, which was extremely low (lower than 3.6%) in the rest of the ingredients presented in treatment 2: two defatted ingredients (canola meal and rice bran defatted), alfalfa pellets, the hulls of peanut and rice, and coffee chaff [26,27,33,34]. The lipid content of corn DDGS and spent brewery DG is reported to be higher than 8% [26,27,30,34,35,37].

The optimal macro nutrient ratios for *T. molitor* may be those observed in the best performing treatments (1, 5, and 8): 0.06 ± 0.03 (max 0.12 min 0.03), 0.23 ± 0.01 (max 0.25 min 0.2), and 0.71 ± 0.03 (max 0.75 min 0.65) for lipid, protein, and carbohydrate, respectively. Rho and Lee (2016) [45] determined that an equal ratio of protein and carbohydrate was the best for *T. molitor* based on adult fecundity and longevity. However, this study is not comparable with ours because both studies were done on different life stages and measured different life cycle parameters.

Ingredients that were considered relevant based on relative consumption percentage (over 10%) included potato, cabbage, wheat bran, crude rice bran, defatted rice bran, corn DDGS, spent brewery DG, canola meal, and sunflower meal. Multiple regression analyses of consumption of relevant ingredients versus live biomass gain showed significant positive effects. These results can be interpreted as evidence that such ingredients are suitable for inclusion in diets for *T. molitor*, especially when biomass production is one of the main priorities. However, consumption of relevant ingredients did not always have positive effects on food assimilation. For instance, canola and sunflower meals had significant negative effects on assimilation. Food assimilation is not necessarily critical for biomass production when the food provided has a low cost, as in this case where agricultural by-products are used. Analysis of nutrient intake ratios showed that intake of fiber negatively affects food assimilation. This may explain the negative effects of canola and sunflower meals on assimilation, since both meals have a relatively high fiber content. Food conversion efficiency (ECI) was impacted positively by the consumption of potato, rice bran and canola meal. Because both food assimilation and ECI significantly impacted biomass gain in a positive way, we may consider the ingredients that impact both parameters in a positive way as highly suitable for inclusion in insect diets. Potato, rice bran, cabbage, spent brewery DG, and canola meal seem to be highly suitable as ingredients in *T. molitor* diets, but defatted rice bran, corn DDGS, and sunflower meal are promising if provided in the correct proportions. Wheat bran, potato, and cabbage have been used and are currently used regularly in *T. molitor* diets for mass production [46]. The rest of the ingredients are not currently used in commercial production, but some studies have assessed their potential, such as on spent brewery DG [32].

Macro-nutrient intake ratios were an important factor affecting live biomass gain, food assimilation and ECI. Macro-nutrient ratios were optimal for *T. molitor* within ranges of 0.06 ± 0.03, 0.23 ± 0.01, and 0.71 ± 0.03 for lipid, protein, and carbohydrate, respectively. Nutrient intake analyses showed that the intake of carbohydrate significantly and positively impacted live biomass gain, food assimilation and ECI. The intake of protein did not impact these three parameters within the ranges observed in this study. It appears that protein intake was strongly regulated by self-selection in most treatments, with the only exception of treatment 2. Other studies have reported that high protein intake reduce development time and pupal size [42] and increased adult longevity and fecundity [43]. In this study the impact of high intake of protein on biomass productivity and food utilization was negative. High intake levels of fiber also had a negative impact on food assimilation and ECI. Li et al. (2015) [47] reported that the optimal intake levels of crude fiber for *T. molitor* is within a range of 5 to 10%. In this study we did not compare intakes of crude fiber, but the self-selected percentages of ND fiber were between 22.52 ± 0.62% in treatment 5 and 34.94 ± 0.94% in treatment 9.

## 5. Conclusions

The macro-nutrient intake ratios resulting from ingredient self-selection by *T. molitor* fell within narrow margins: Lipid intake was between 0.12 and 0.03, protein between 0.25 and 0.2, and carbohydrate between 0.75 and 0.65. Deviations from these ranges of macro nutrient intake ratios resulted in a diminished performance in larval growth and food utilization.

The relevant ingredients, based on their relative consumption by *T. molitor* larvae included potato, cabbage, wheat bran, crude rice bran, defatted rice bran, corn DDGS, spent brewery DG, canola meal, and sunflower meal. Consumption of relevant ingredients significantly affected live biomass production in a positive way in *T. molitor* larvae.

Both food assimilation and efficiency of conversion of ingested food were positively impacted by ingestion of carbohydrate and negatively impacted by ingestion of fiber. Ingredients that enhanced both of these parameters had relatively high carbohydrate and low fiber content such as potato. However, levels of carbohydrate and fiber should not depart from the self-selected ranges observed, because excessive or deficient intake of those nutrients can have a detrimental impact on growth and food utilization in *T. molitor* larvae.

## Figures and Tables

**Figure 1 insects-11-00827-f001:**
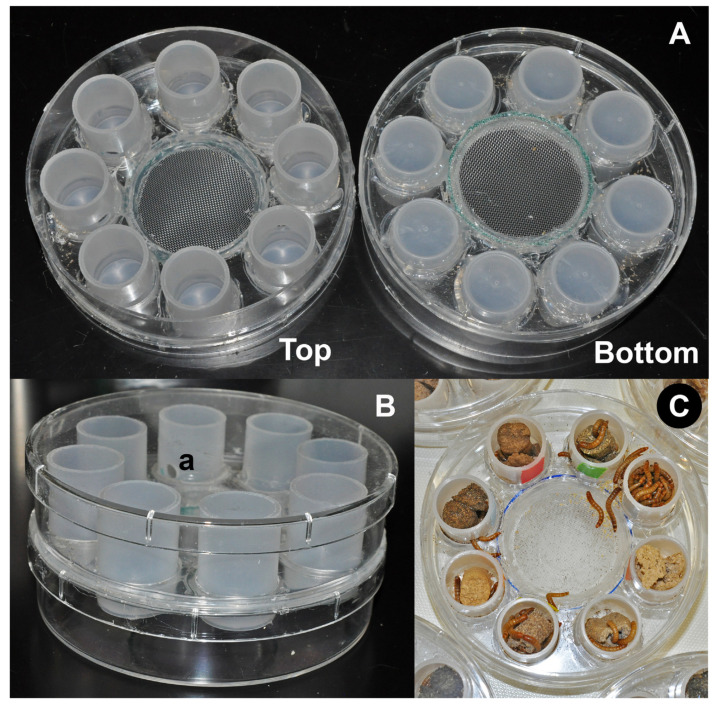
Multiple-choice arenas for self-selection experiments: (**A**) top and bottom views; (**B**) assembled arena with cover and bottom dish; (**C**) experimental unit with food choices and larvae, (a) opening into modified vial as food-choice compartment.

**Figure 2 insects-11-00827-f002:**
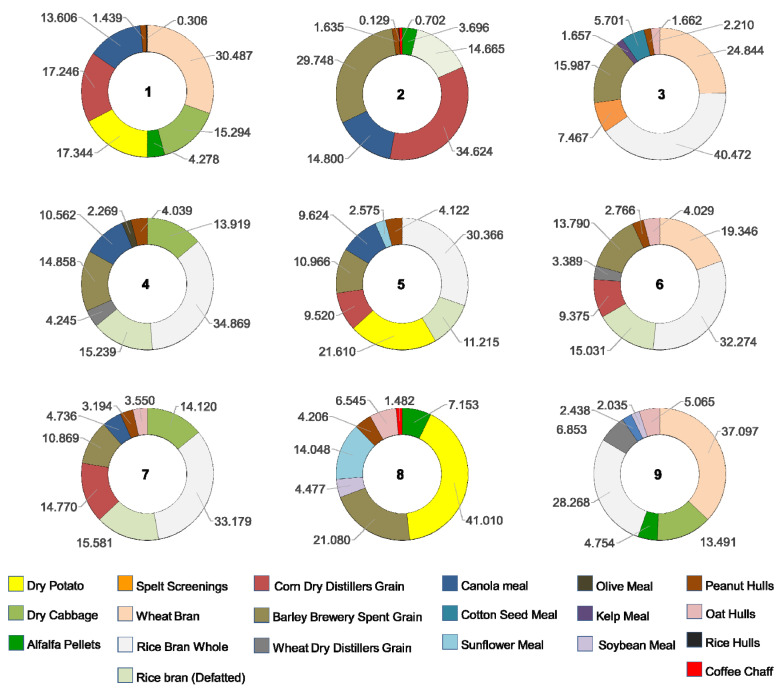
Proportional consumption of food ingredients by *T. molitor* larvae in nine treatments of different combinations of eight ingredients.

**Figure 3 insects-11-00827-f003:**
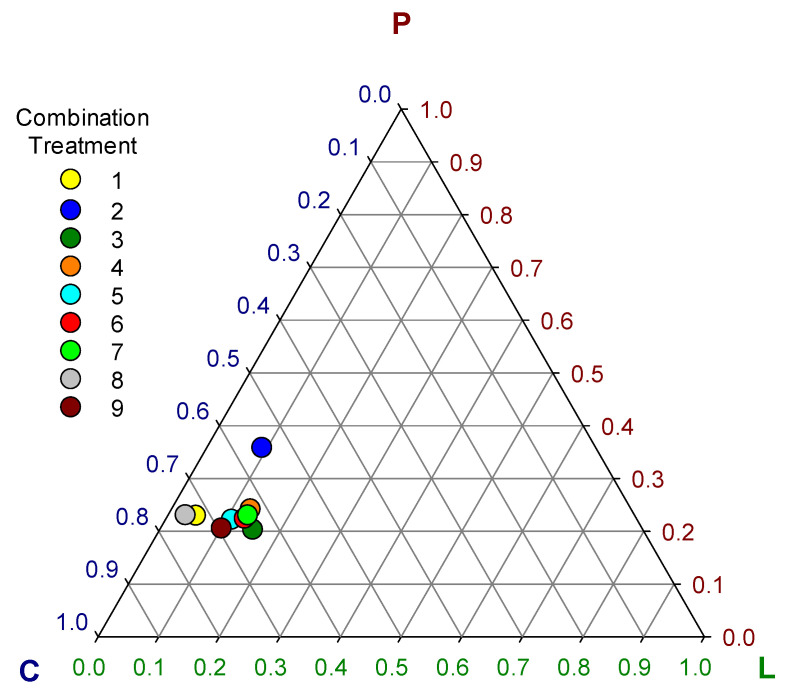
Ternary plot of self-selected macro-nutrient intake ratios (L = lipid, P = protein, C = carbohydrate) in nine combination treatments of eight ingredients.

**Figure 4 insects-11-00827-f004:**
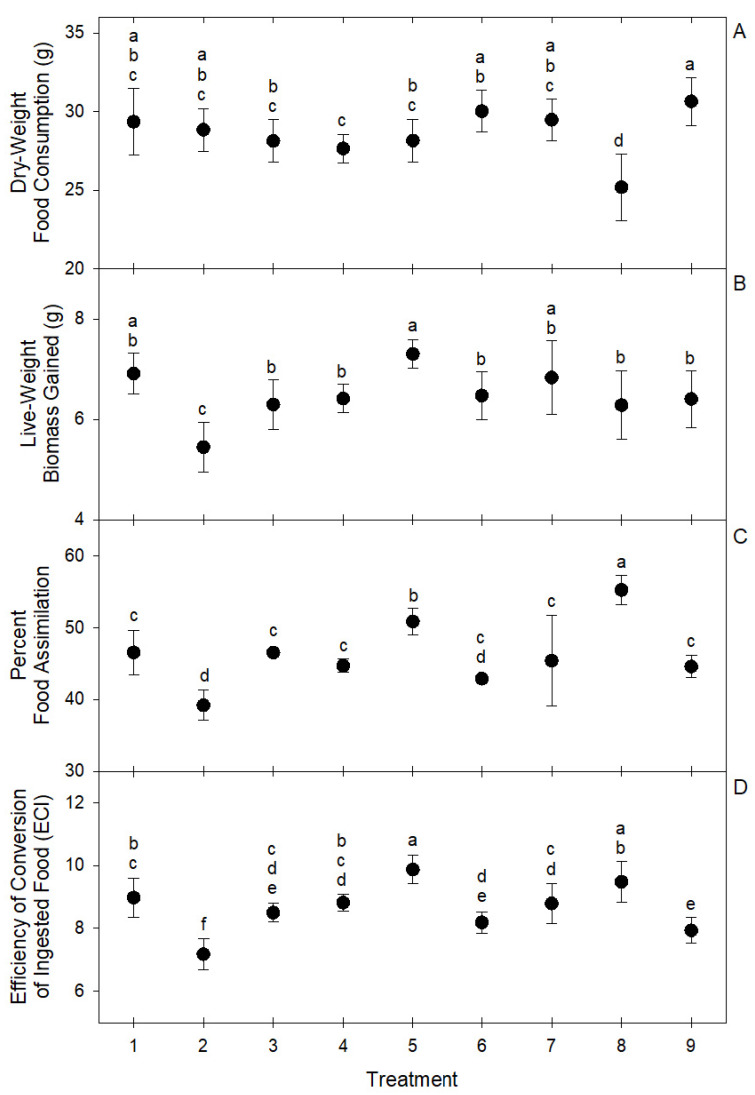
Circles represent means and brackets represent standard deviation of (**A**) dry-weight food consumption, (**B**) live biomass gain, (**C**) percent food assimilation, and (**D**) efficiency of conversion of ingested food (ECI) by groups of 100 *T. molitor* larvae in nine self-selection treatments of eight ingredients. Means with the same letter are not significantly different at α = 0.05 after Tukey–Kramer HSD test.

**Table 1 insects-11-00827-t001:** Combination of eight food choices presented to *Tenebrio molitor* larvae groups in nine self-selection treatments.

Ingredient	Treatment								
	1	2	3	4	5	6	7	8	9
White Cabbage	X			X			X		X
Potato	X				X			X	
Alfalfa pellets	X	X						X	X
Wheat Bran	X		X			X			X
Rice bran whole			X	X	X	X	X		X
Rice bran defatted		X		X	X	X	X		
Spelt Screenings			X						
Corn dry distiller’s grain	X	X			X	X	X		
Wheat dry distiller’s grain				X		X			X
Barley brewery spent grain		X	X	X	X	X	X	X	
Canola meal	X	X		X	X		X		X
Soy Meal								X	X
Sunflower meal					X			X	
Olive meal				X					
Cotton seed meal			X						
Kelp meal			X						
Oat hulls			X			X	X	X	X
Peanut hulls	X	X	X	X	X	X	X	X	
Rice hulls	X	X							
Coffee chaff		X						X	

**Table 2 insects-11-00827-t002:** Dry-weight consumption (g) of food ingredients by *T. molitor* larvae in nine self-selection treatment combinations of eight choices during a three-week period.

Ingredient	Treatment								
	1	2	3	4	5	6	7	8	9
White Cabbage	4.49 ± 0.91			3.85 ± 0.35			4.16 ± 0.61		4.14 ± 0.45
Potato	5.09 ± 0.63				6.09 ± 0.55			10.33 ± 0.93	
Alfalfa pellets	1.26 ± 0.22	1.07 ± 0.24						1.8 ± 0.33	1.46 ± 0.24
Wheat Bran	8.95 ± 1.1		6.99 ± 0.98			5.81 ± 0.79			11.37 ± 0.81
Rice bran whole			11.38 ± 0.98	9.64 ± 0.73	8.55 ± 0.88	9.69 ± 0.51	9.78 ± 2.09		8.67 ± 1.7
Rice bran defatted		4.23 ± 1.19		4.21 ± 0.72	3.16 ± 1.03	4.51 ± 0.89	4.59 ± 1.84		
Spelt Screenings			2.1 ± 0.35						
Corn DDGS	5.06 ± 1.09	9.99 ± 1.23			2.68 ± 0.65	2.82 ± 0.64	4.35 ± 0.82		
Wheat DDGS				1.17 ± 0.26		1.02 ± 0.16			2.1 ± 0.75
Barley brewery spent grain		8.58 ± 0.59	4.5 ± 0.59	4.11 ± 0.52	3.09 ± 0.89	4.14 ± 0.4	3.2 ± 0.43	5.31 ± 0.43	
Canola meal	3.99 ± 0.9	4.27 ± 0.85		2.92 ± 0.31	2.71 ± 0.3		1.4 ± 0.46		0.75 ± 0.35
Soybean Meal								1.13 ± 0.14	0.62 ± 0.07
Sunflower meal					0.73 ± 0.16			3.54 ± 1.03	
Olive meal				0.63 ± 0.11					
Cotton seed meal			1.6 ± 0.35						
Kelp meal			0.47 ± 0.05						
Oat hulls			0.62 ± 0.29			1.21 ± 0.22	1.05 ± 0.19	1.65 ± 0.36	1.55 ± 0.39
Peanut hulls	0.42 ± 0.19	0.47 ± 0.12	0.47 ± 0.18	1.12 ± 0.21	1.16 ± 0.31	0.83 ± 0.18	0.94 ± 0.38	1.06 ± 0.54	
Rice hulls	0.09 ± 0.03	0.04 ± 0.02							
Coffee shaft		0.2 ± 0.09						0.37 ± 0.14	
Total Consumption	29.36 ± 2.13	28.84 ± 1.36	28.13 ± 1.36	27.06 ± 0.9	28.16 ± 1.35	30.03 ± 1.32	29.48 ± 1.35	25.18 ± 2.12	30.66 ± 1.53

Mean ± standard deviation.

**Table 3 insects-11-00827-t003:** Macro nutrient intake ratios of *T. molitor* larvae in nine self-selection treatments with different combinations of eight food ingredients.

Treatment	Lipid	Protein	Carbohydrate
1	0.047 ± 0.003 ^e^	0.23 ± 0.007 ^c^	0.723 ± 0.008 ^b^
2	0.092 ± 0.003 ^d^	0.359 ± 0.006 ^a^	0.549 ± 0.008 ^g^
3	0.157 ± 0.006 ^a^	0.206 ± 0.005 ^d^	0.637 ± 0.003 ^e^
4	0.132 ± 0.006 ^b^	0.244 ± 0.005 ^b^	0.624 ± 0.006 ^f^
5	0.112 ± 0.005 ^c^	0.22 ± 0.012 ^c^	0.668 ± 0.01 ^d^
6	0.134 ± 0.004 ^b^	0.23 ± 0.004 ^c^	0.636 ± 0.005 ^e^
7	0.132 ± 0.017 ^b^	0.23 ± 0.008 ^c^	0.638 ± 0.014 ^e^
8	0.029 ± 0.002 ^f^	0.23 ± 0.008 ^c^	0.741 ± 0.009 ^a^
9	0.101 ± 0.011 ^d^	0.205 ± 0.009 ^d^	0.694 ± 0.009 ^c^

Mean ± standard deviation. Means with the same letter are not significantly different after Tukey–Kramer HSD test at α = 0.05.

**Table 4 insects-11-00827-t004:** Summarized estimated nutrient intake means and ranges in 90 self-selection observations from 9 different 8-choice combination treatments (in mg/100 mg).

Nutrient	Mean ± SD	Minimum	Maximum
Lipid	8.13 ± 3.13	2.27	13.03
Protein	18.75 ± 2.78	15.58	26.45
Carbohydrate	51.94 ± 6.54	38.71	64.84
Fiber (ND)	28.06 ± 3.48	21.72	36.41
Ca	0.445 ± 0.246	0.11	0.906
Fe	0.014 ± 0.002	0.009	0.018
Mg	0.439 ± 0.113	0.157	0.568
Zn	0.004 ± 0.0007	0.002	0.005
Mn	0.007 ± 0.002	0.002	0.01

**Table 5 insects-11-00827-t005:** Model from stepwise on percent assimilation.

Parameter	Estimate	Sum of Squares	*F* Ratio	*p* > *F*
Carbohydrate	5.4 × 10^−4^ ± 4.3 × 10^−5^	1121.48	158.76	<0.0001
Fiber	−7.4 × 10^−4^ ± 8.1 × 10^−5^	584.35	82.72	<0.0001

Model: R^2^ = 0.731; F = 117.93; df 2, 87; *p* < 0.0001.

**Table 6 insects-11-00827-t006:** Model from Stepwise and backwards elimination on ECI.

Parameter	Estimate	Sum of Squares	*F* Ratio	*p* > *F*
Carbohydrate	1.2 × 10^−4^ ± 1.0 × 10^−5^	27.22	120.17	<0.0001
Fiber	−1.6 × 10^−4^ ± 1.5 × 10^−5^	24.88	109.83	<0.0001
Mg	0.017 ± 0.003	6.14	27.1	<0.0001
Mn	−0.667 ± 0.14	4.14	22.71	<0.0001

Model: R^2^ = 0.73; F = 57.48; df 4, 85; *p* < 0.0001.

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
