# Peer review of "Self-Selection of Agricultural By-Products and Food Ingredients by Tenebrio molitor (Coleoptera: Tenebrionidae) and Impact on Food Utilization and Nutrient Intake"

_insects, 2020, doi:10.3390/insects11120827_

Round 1
Reviewer 1 Report
- Line 15: Change text “by products” to “by-products”.
- Abstract: Please avoid using abbreviations in the abstract (RH, ECI, NDF, DDGS)
- Materials and methods section: Clearly describe the number of larva in each plastic dish. Was it 10 per dish or some other number?
- Materials and methods section: Were there dead larva? What was the mortality percentage? How were the dead taken into consideration for the various measurements (FC, ECI, etc)?
- Line 70: The “Morales-Ramos et al.” reference has both the year (2019) and the number of the reference [22]. According to the Journal’s Instruction for Authors references only reference number should be used. Therefore, the year should be deleted. Please make this change in all other similar cases in the text (lines 73, 142, 144, 145, 148, etc)
- Line 129: Change “remining” to “remaining”.
- Line 132: Change “remining” to “remaining”.
- Results section: In the results section the authors should greatly limit the amount of numbers presented, such as F and data values & St.Deviation. F is not needed if P is shown. In addition, data values and SD already appear in the tables and figures, and they are just repeated there.
- Lines 352-358: Check if this section is needed and revise it accordingly.
- Tables: Please use the same number of decimals for all values (for example in table 3, treatment 1, write protein as 0.230). Also, the “letters” that signify differences should be added as superscript after the mean values, not after SD.
- Figure 2: Change “Rice bran hole” to “Rice bran whole”.
- Figure 3: At first reading, it is quite confusing to identify which side of the triangle corresponds to each nutrient. I would suggest that the authors add the nutrient in the middle under each triangle side. For example, under the bottom side of the triangle they write “protein”. Another idea is to write the nutrient with the same color as the numbers of the side it corresponds (for example write Protein and bottom side numbers with the same color letters).
Author Response
Responses to Reviewer 1:
- Line 15: Change text “by products” to “by-products”.
Done
- Absract: Please avoid using abbreviations in the absract (RH, ECI, NDF, DDGS)
Except for RH, all the abbreviations have been removed from the Abstract.
- Materials and methods section: Clearly describe the number of larva in each plasic dish. Was it 10 per dish or some other number?
This is clearly stated in line 77 “Experimental units consisted of groups of 100 larvae maintained in multiple-choice arenas designed to provide equal access to 8 different food choices.”. Further clarification has been added in line 119.
- Materials and methods section: Were there dead larva? What was the mortality percentage? How were the dead taken into consideration for the various measurements (FC, ECI, etc)?
A paragraph was added for clarification on how mortality was accounted for.
- Line 70: The “Morales-Ramos et al.” reference has both the year (2019) and the number of the reference [22]. According to the Journal’s Insruction for Authors references only reference number should be used. Therefore, the year should be deleted. Please make this change in all other similar cases in the text (lines 73, 142, 144, 145, 148, etc)
Corrected
- Line 129: Change “remining” to “remaining”.
Corrected
- Line 132: Change “remining” to “remaining”.
Corrected
- Results section: In the results section the authors should greatly limit the amount of numbers presented, such as F and data values & St.Deviation. F is not needed if P is shown. In addition, data values and SD already appear in the tables and fgures, and they are jus repeated there.
Reporting F statistic in addition to P in the text of “Results” is standard in most journals. The same can be said for the standard deviations next to the means. This is required for most journals and we respectfully disagree with the reviewer in this point. We will retain them unless specifically directed by the editor to remove them.
- Lines 352-358: Check if this section is needed and revise it accordingly.
Deleted
- Tables: Please use the same number of decimals for all values (for example in table 3, treatment 1, write protein as 0.230). Also, the “letters” that signify diferences should be added as superscript after the mean values, not after SD.
This is a similar case as the use of F and SD in the text. It is standard in most journals to place the letters that signify significant differences after the standard deviations. We think that placing them next to the means and before the “±” symbol would be confusing. We respectfully request to leave them unchanged.
- Figure 2: Change “Rice bran hole” to “Rice bran whole”.
Corrected
- Figure 3: At frs reading, it is quite confusing to identify which side of the triangle corresponds to each nutrient. I would suggest that the authors add the nutrient in the middle under each triangle side. For example, under the bottom side of the triangle they write “protein”. Another idea is to write the nutrient with the same color as the numbers of the side it corresponds (for example write Protein and bottom side numbers with the same color letters).
We think that this is an excellent idea, and we changed the color of the axes to match the titles color. This seems to solve the problem of confusion that some readers experience with this type of plots. Labels are in the extremes because they mark the points of highest value and facilitates the interpretation of the plot by providing points of reference for the reader. The dots distance to the labels can be directly interpreted as the relative content of each macro-nutrient.

Reviewer 2 Report
Dear Author,
I read your interesting manuscript entitled “Self-Selection of Agricultural By-Products and Food Ingredients by Tenebrio molitor (Coleoptera: Tenebrionidae) and Impact on Food Utilization and Nutrient Intake”. The paper it is weel written and original. I think it only needs some minor revision that I list below:
Line 32: replace weigh with weight.
Line 62 and throughout the text: check if the year of publication has to be removed.
Line 68: Materials and methods. I suggest to divide this long section in subparagraph (e.g. Colony, Experimental diets…….).
Line 127: How did you separate the food from the frass?
Line 144-145: please correct the references, there is a repetition.
Line 181 and throughout the text (also in the figure captation): please insert if it is a percentage or gram.
Line 264: Please replace 42 with 43, there is a repetition.
Line 312: replace shower with showed.
Author Response
Responses to Reviewer 2:
Dear Author,
I read your interesing manuscript entitled “Self-Selection of Agricultural By-Products and Food Ingredients by Tenebrio molitor (Coleoptera: Tenebrionidae) and Impact on Food Utilization and Nutrient Intake”. The paper it is weel written and original. I think it only needs some minor revision that I lis below:
Line 32: replace weigh with weight.
Corrected
Line 62 and throughout the text: check if the year of publication has to be removed.
Corrected
Line 68: Materials and methods. I sugges to divide this long section in subparagraph (e.g. Colony, Experimental diets…….).
The Materials and Methods section was divided in three subsections.
Line 127: How did you separate the food from the frass?
This was briefly explained in lines 130-131. A paragraph has been added to clarify this explanation further.
Line 144-145: please correct the references, there is a repetition.
Corrected
Line 181 and throughout the text (also in the fgure captation): please insert if it is a percentage or gram.
These values are 3-way ratios. A paragraph was added to explain how these ratios were calculated.
Line 264: Please replace 42 with 43, there is a repetition.
Corrected
Line 312: replace shower with showed.
Corrected
